# LoraQuant: Mixed-Precision Quantization of LoRA to Ultra-Low Bits for LLM Customization

**Amir Reza Mirzaei**[1*]                                    *amirreza2001@gmail.com*
**Yuqiao Wen**[1*]                                                  *yq.when@gmail.com*
**Yanshuai Cao**[2]                                     *yanshuai.cao@borealisai.com*
**Lili Mou**[1,3]                                        *doublepower.mou@gmail.com*

[1] *Department of Computing Science and Alberta Machine Intelligence Institute (Amii), University of Alberta*
[2] *RBC Borealis*
[3] *Canada CIFAR AI Chair*

**Reviewed on OpenReview:** *https://openreview.net/forum?id=71svCWi178*

## Abstract

Low-Rank Adaptation (LoRA) has become a popular technique for parameter-efficient fine-tuning of large language models (LLMs). In many real-world scenarios, multiple adapters are loaded simultaneously to enable LLM customization for personalized user experiences or to support a diverse range of tasks. Although each adapter is lightweight in isolation, their aggregate cost becomes substantial at scale. To address this, we propose LoRaQuant, a mixed-precision post-training quantization method tailored to LoRA. Specifically, LoRaQuant reparameterizes each adapter by singular value decomposition (SVD) to concentrate the most important information into specific rows and columns. This makes it possible to quantize the important components to higher precision, while quantizing the rest to lower bitwidth. We conduct comprehensive experiments with LLaMA 2-7B, LLaMA 2-13B, and Mistral 7B models on mathematical reasoning, coding, and summarization tasks. Results show that our LoRaQuant uses significantly lower bits than other quantization methods, but achieves comparable or even higher performance.[1]

## 1 Introduction

Large Language Models (LLMs) have achieved remarkable performance across a wide range of natural language tasks (Ouyang et al., 2022; Wang et al., 2022; Zhao et al., 2023), but fine-tuning LLMs for new applications remains computationally and memory intensive. To address this challenge, low-rank adaptation (LoRA; Hu et al., 2022) has emerged as a widely adopted method for parameter-efficient fine-tuning. LoRA introduces small, task-specific low-rank matrices, and during the adaptation, only these low-rank matrices are trained while the base model is frozen.

An increasingly important use case of LoRA is LLM customization, as LLM providers (e.g., OpenAI and Google) often allow users to personalize their own LLMs (OpenAI, 2025; Google Cloud). This could result in hundreds of millions of customized LLMs addressing diverse tasks, domains, and users. This imposes significant challenges of storing and using these massive customized LLMs.

A straightforward attempt to solve these challenges is to freeze the base LLM and train a separate adapter for each customization (Sheng et al., 2024). During deployment, multiple LoRAs are often loaded simultaneously due to parallel user requests, and thus, the memory footprint of LoRAs becomes a concern, especially if the GPU memory is small. This is because, although each individual LoRA is relatively lightweight, the cumulative GPU memory consumption for loading many adapters can become significant.

---

[*]Project partially done during Mitacs internship at RBC Borealis.
[1]Our code is released at: **https://github.com/MANGA-UOFA/LoRAQuant**

To scale multi-LoRA systems, Gabrielsson et al. (2024) propose a compression technique that clusters different LoRAs and enables representation sharing within each cluster. However, a key limitation of their method is that it requires recomputing the shared parameters whenever a new adapter is added. Additionally, their evaluation is primarily conducted on relatively simple tasks where the base LLM already performs well and their approach struggles on more challenging tasks (see Appendix F).

In this paper, we propose LORAQUANT, a post-training LoRA quantization method for LLM customization. Quantization is a well-established technique for compressing neural networks, and is able to substantially reduce the parameter space without degrading performance much (Frantar et al., 2023; Lin et al., 2024). Despite this progress, the quantization of LoRA has received little attention in the literature, compared with the quantization of full LLMs. Although existing quantization methods can be directly applied to LoRA weights, they overlook the unique low-rank structure of LoRA and perform poorly at ultra-low precisions (e.g., 1–2 bits).

In our work, we observe that a LoRA is a product of two low-rank matrices, which can be easily split into multiple *lower*-rank adapters (sub-LoRAs). By reparametrizing the LoRA with singular value decomposition (SVD), we can perform mixed-precision quantization for different sub-LoRAs: more precisions for more important SVD dimensions and fewer precisions for less important ones. With our approach, we are able to retain high performance of LoRA while reducing the memory space to a large extent.

In the experiments, we evaluate our method by training adapters on three representative models, LLaMA2-7B, LLaMA2-13B (Touvron et al., 2023), and Mistral-7B (Jiang et al., 2023), across diverse tasks including mathematical reasoning, code generation, and summarization. Our results demonstrate that LORAQUANT achieves competitive performance even under ultra-low bitwidth for LoRA (fewer than 2 bits on average).

## 2 Related Work

**Model quantization.** Quantization has become an important technique for reducing the memory footprint of LLMs, enabling efficient deployment without a significant loss in accuracy. A variety of post-training methods have been proposed to compress full-precision weights into lower-bit representations. For instance, GPTQ (Frantar et al., 2023) leverages second-order information to minimize quantization error, while AWQ (Lin et al., 2024) incorporates activation statistics to guide weight quantization. InvarExplore (Wen et al., 2025) leverages model invariances (namely, rotation, scaling, and permutation) to make weights perform better after quantization. SVDQuant (Li et al., 2025) performs quantization on an entire weight matrix, but adds a full-precision SVD with a few dimensions to improve performance. Our work differs from SVDQuant (although also using SVD) in that we focus on LoRA quantization and use the SVD decomposition to split the LoRA into two sub-LoRAs: one containing more important information and the other less important information. For a more detailed discussion, see Appendix E.

Although the above approaches primarily target moderate quantization (e.g., 3–8 bits), an even more extreme direction is binarization, where weights are restricted to two values (Rastegari et al., 2016). Pure binary quantization usually achieves very low performance, and researchers have proposed mixed-precision methods where some weights are binarized while others are kept in high precision. For example, PB-LLM (Shang et al., 2024) uses an additional bit to indicate whether a weight is binarized or not, which unfortunately offsets the memory saved. BiLLM (Huang et al., 2024) also adopts mixed-precision quantization, but restricts the binarization to certain column of the weight matrix. However, BiLLM still requires an additional indicator bit; it employs a split binarization strategy in which the weights are divided into two groups and binarized separately, so the extra bit is needed to indicate the group membership of each weight.

Another branch of methods for achieving ultra-low precision is quantization-aware training (QAT). Unlike post-training quantization, QAT incorporates quantization during training so that the model can adapt to low-bit constraints. For example, BitNet (Wang et al., 2025) trains models directly at low precision. Since quantization operations are non-differentiable, such approaches typically rely on the straight-through estimator (STE) to approximate gradients during backpropagation. We do not focus on QAT methods in this

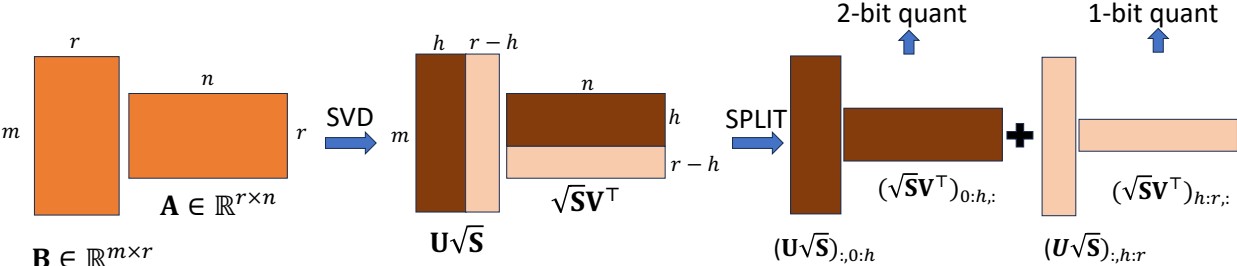

Figure 1: Overview of our LoraQuant method.

work, as they generally require full model training from scratch and therefore incur substantial computational cost (Gholami et al., 2021).

**Low-rank adapter (LoRA).** LoRA (Hu et al., 2022) has become a widely adopted approach for parameter-efficient fine-tuning of LLMs. Building on this idea, several extensions aim to further reduce memory overhead or improve training effectiveness. Meng et al. (2024) initialize LoRA with the SVD of the base weight matrix rather than random values, providing a stronger starting point for optimization. Zhang et al. (2023) dynamically adjust the rank of LoRA during training, allocating higher ranks to more important layers. Hao et al. (2024) and Lialin et al. (2024) address the limitation that LoRA updates may remain low-rank by iteratively merging and resampling adapters during training. Kopiczko et al. (2024) propose sharing LoRA weights across layers to reduce memory.

LoRA has also been applied to fine-tuning quantized models. QLoRA (Dettmers et al., 2023) freezes the quantized base model while training an add-on LoRA in full precision. LoftQ (Li et al., 2024) and ApiQ (Liao et al., 2024) improve QLoRA adapter initialization by choosing parameters that reduce quantization errors instead of random values. QA-LoRA (Xu et al., 2024) extends QLoRA by ensuring that the adapter weights remain easily quantizable even after being merged with the original weights at the cost of reducing the representational capacity of the adapters during training.

Note that our LoraQuant is completely different from QLoRA and its variants, despite similar names. QLoRA uses a full-precision LoRA to fine-tune a quantized model, as the latter cannot be easily trained. Our work focuses on the post-training quantization of LoRAs. It is arguable that LoRA is already a parameter-efficient method, but with increasingly many LoRAs for LLM customization, the aggregated cost can be substantial.

In the context of serving multiple LoRAs simultaneously, prior work has largely focused on hardware-oriented implementation. S-LoRA (Sheng et al., 2024) proposes a batched inference strategy to improve throughput when handling concurrent LoRA requests. Punica (Chen et al., 2024) implements a customized GPU kernel, Segmented Gather Matrix–Vector Multiplication (SGMV), which enables efficient batching across heterogeneous LoRAs. Complementary to these approaches, Gabrielsson et al. (2024) reduce the memory footprint by weight sharing among a cluster of LoRAs and assigning a few additional parameters for each LoRA. However, our experiments will show that this approach does not perform well in sophisticated tasks such as math reasoning and coding (§4.2).

## 3 Our LoraQuant Method

In our LoraQuant method, we decompose a LoRA into two lower-rank adapters, called sub-LoRAs (§3.1). Then, we allocate slightly higher precision to the more important sub-LoRA, and perform extreme 1-bit quantization for the less important sub-LoRA (§3.2). To further mitigate quantization error, we also apply gradient-based optimization before quantization (§3.3). An overview of the approach is shown in Fig. 1, and the step-by-step procedure is given in Algs. 1 and 2.

### 3.1 Splitting a LoRA into Sub-LoRAs by SVD

For an update of neural network weights $\mathbf{W} \leftarrow \mathbf{W} + \Delta\mathbf{W}$, a low-rank adapter (LoRA; Hu et al., 2022) learns two low-rank matrices $\mathbf{B} \in \mathbb{R}^{m \times r}$ and $\mathbf{A} \in \mathbb{R}^{r \times n}$ to approximate $\Delta\mathbf{W}$, where $r \ll \min(m, n)$. In other words, the update becomes $\mathbf{W} \leftarrow \mathbf{W} + \mathbf{BA}$.

To enable mixed-precision quantization, we split the LoRA into sub-LoRAs, and in particular, we have two sub-LoRAs in our experiments.[2] In other words, we need to find sub-LoRAs, namely, $\mathbf{B}^{(h)}\mathbf{A}^{(h)}$ and $\mathbf{B}^{(l)}\mathbf{A}^{(l)}$, such that $\mathbf{B}^{(h)}\mathbf{A}^{(h)} + \mathbf{B}^{(l)}\mathbf{A}^{(l)} = \mathbf{BA}$. Later, $\mathbf{B}^{(h)}$ and $\mathbf{A}^{(h)}$ will be quantized with higher precisions, whereas $\mathbf{B}^{(l)}$ and $\mathbf{A}^{(l)}$ will be quantized with lower precisions.

The key challenge is to determine how to perform the split in a way that best suits mixed-precision quantization. Intuitively, the more important components should be allocated to the higher-precision sub-LoRA, while the less important components can be assigned to the lower-precision sub-LoRA. A naïve strategy is to split $\mathbf{B}$ and $\mathbf{A}$ by selecting certain columns and rows based on weight norms or quantization error. However, information is usually scattered over rows and columns of neural weights, so such an approach is less effective.

Instead, we propose to reparameterize a low-rank adapter $\mathbf{BA}$ into an equivalent factorization $\mathbf{B}'\mathbf{A}'$ such that $\mathbf{BA} = \mathbf{B}'\mathbf{A}'$, where important information is concentrated in specific rows and columns of $\mathbf{B}'$ and $\mathbf{A}'$. This will better suit our mixed-precision quantization scheme.

To accomplish this, we apply the singular value decomposition (SVD) to the original adapter $\mathbf{BA}$:

$$\mathbf{BA} = \mathbf{USV}^\top, \tag{1}$$

where $\mathbf{U} \in \mathbb{R}^{m \times r}$ and $\mathbf{V} \in \mathbb{R}^{n \times r}$ are orthonormal matrices, and $\mathbf{S} \in \mathbb{R}^{r \times r}$ is a diagonal matrix containing the singular values in descending order. Here, the SVD is truncated to $r$ ranks, since $\mathbf{BA}$ has at most $r$ ranks. We reparameterize the LoRA by

$$\mathbf{B}' = \mathbf{US}^{1/2}, \quad \mathbf{A}' = \mathbf{S}^{1/2}\mathbf{V}^\top, \tag{2}$$

where $\mathbf{S}^{1/2}$ is the diagonal matrix whose entries are the square roots of the singular values. It is easy to verify $\mathbf{B}'\mathbf{A}' = (\mathbf{US}^{1/2})(\mathbf{S}^{1/2}\mathbf{V}^\top) = \mathbf{USV}^\top = \mathbf{BA}$.

This reparameterization ranks the importance of each component (i.e., each column of $\mathbf{B}'$ and corresponding row of $\mathbf{A}'$) by the magnitude of its associated singular value. We retain the top-$h$ components in higher precision and quantize the remaining $r - h$ components using a lower bitwidth.

Formally, the more important sub-LoRA is given by:

$$\mathbf{B}^{(h)} = (\mathbf{US}^{1/2})_{[:,\,0:h]} \quad \mathbf{A}^{(h)} = (\mathbf{S}^{1/2}\mathbf{V}^\top)_{[0:h,\,:]} \tag{3}$$

and the less important sub-LoRA is given by:

$$\mathbf{B}^{(l)} = (\mathbf{US}^{1/2})_{[:,\,h:r]}, \quad \mathbf{A}^{(l)} = (\mathbf{S}^{1/2}\mathbf{V}^\top)_{[h:r,\,:]}, \tag{4}$$

where the subscript $[\cdot, \cdot]$ chooses certain columns and rows. We can easily see that $\mathbf{B}^{(h)}\mathbf{A}^{(h)} + \mathbf{B}^{(l)}\mathbf{A}^{(l)} = \mathbf{B}'\mathbf{A}' = \mathbf{BA}$, showing that our transformations do not change the functionality of the LoRA.

**Determining the ratio of two sub-LoRAs.** We employ a dynamic strategy to compute $h$ for each adapter in a model, based on the coverage of total variance of that adapter. Specifically, we introduce a ratio hyperparameter $\rho \in (0, 1]$, which specifies the minimum ratio of total variance that must be preserved. Given the singular values $s_1, s_2, \cdots, s_r$ (sorted from largest to smallest) of an adapter, we compute $h$ as the smallest integer satisfying

$$\frac{\sum_{i=1}^{h} s_i^2}{\sum_{i=1}^{r} s_i^2} \geq \rho. \tag{5}$$

---

[2]Our framework can be easily extended to more than two groups with different bit allocations. However, in our experiments, two groups already achieve strong performance, so we adopt this simpler setting.

This ensures that the top-$h$ singular directions collectively explain at least $\rho \times 100\%$ of the variance in the adapter product. Compared with directly setting $h$ as a hyperparameter, our strategy allows us to adaptively allocate more precision to layers that require a greater number of singular components to preserve their representational capacity.

## 3.2 Quantization Methods

After splitting a LoRA into two sub-LoRAs, we perform mixed-precision quantization. For the more important sub-LoRA $\mathbf{B}^{(h)}\mathbf{A}^{(h)}$, we use round-to-nearest quantization (RTN; Jacob et al., 2018) with a higher precision (e.g., 2 bits for a weight). For the less important sub-LoRA $\mathbf{B}^{(l)}\mathbf{A}^{(l)}$, we use binary quantization with only one bit for extreme compression. Profoundly, our mixed-precision quantization with 2 bits and 1 bit allows us to achieve less than 2 bits on average, which is a setting where existing methods cannot perform well (Shang et al., 2024; Huang et al., 2024).

In the rest of this part, we present RTN and binary quantization methods in detail. Notice that the quantization applies to both $\mathbf{A}^{(\cdot)}$ and $\mathbf{B}^{(\cdot)}$, where the subscript $(\cdot)$ refers to either (h) or (l). We take $\mathbf{A}^{(\cdot)}$ as an example in the following presentation.

**RTN quantization for the more important sub-LoRA.** We employ the widely used round-to-nearest (RTN) method to quantize $\mathbf{B}^{(h)}$ and $\mathbf{A}^{(h)}$. Let us consider $\mathbf{A}^{(h)}$. RTN maps each real-valued weight to the closest integer value, but with a scaling factor $S$ and a zero-point offset $Z$ to cover the range of the weight values. Formally, the integer weights after quantization, denoted by $\bar{\mathbf{A}}^{(h)}$, are

$$\bar{\mathbf{A}}^{(h)} = Q_{\mathrm{RTN}}(\mathbf{A}^{(h)}) = \mathrm{round}\left(\frac{\mathbf{A}^{(h)}}{S}\right) + Z. \tag{6}$$

In RTN, the largest real value in $\mathbf{A}$ is mapped to the largest representable integer, and the smallest value is mapped to the smallest representable integer. Based on this, $S$ and $Z$ are given by

$$S = \frac{\max(\mathbf{A}^{(h)}) - \min(\mathbf{A}^{(h)})}{q_{\max} - q_{\min}}, \qquad Z = \mathrm{round}\left(q_{\min} - \frac{\min(\mathbf{A}^{(h)})}{S}\right), \tag{7}$$

where $q_{\min}$ and $q_{\max}$ denote the minimum and maximum integers representable under the chosen bitwidth. When using the weights, we perform dequantization as

$$D_{\mathrm{RTN}}(\bar{\mathbf{A}}^{(h)}) = S \cdot (\bar{\mathbf{A}}^{(h)} - Z). \tag{8}$$

In implementation, we apply group-wise quantization (Jacob et al., 2018), i.e., the above procedure is performed on a group of contiguous weights instead of the entire matrix. At the cost of introducing more scaling and offset values for fine-grained treatment, this reduces quantization error and improves performance.

**Binary quantization for the less important sub-LoRA.** We perform binary quantization on the less important sub-LoRA $\mathbf{B}^{(l)}\mathbf{A}^{(l)}$ for extreme compression. The classic RTN method is unsuitable in the 1-bit setting, as it maps weights to either $\{0, +S\}$ or $\{0, -S\}$ and cause many weights to collapse to zero due to Eqn. (6), during which significant information is lost.

Instead, we adopt the binary quantization method in Rastegari et al. (2016), which maps values to $\{-S, +S\}$ so as to preserve more representational capacity. In other words, the quantization and dequantization processes are simply given by

$$\bar{\mathbf{A}}^{(l)} = Q_{\mathrm{bin}}(\mathbf{A}^{(l)}) = \mathrm{sign}(\mathbf{A}^{(l)}), \qquad D_{\mathrm{bin}}(\bar{\mathbf{A}}^{(l)}) = S \cdot \bar{\mathbf{A}}^{(l)}, \tag{9}$$

where $\mathrm{sign}(x) = 1$ if $x \geq 0$, or $-1$ otherwise. The scaling factor $S$ is set as

$$S = \frac{1}{l \cdot n} \sum_{i=1}^{l} \sum_{j=1}^{n} \left| A_{ij}^{(l)} \right|, \tag{10}$$

which is shown to minimize the Frobenius norm between the original weights $\mathbf{A}^{(l)}$ and the reconstructed one $D_{\mathrm{bin}}(\bar{\mathbf{A}}^{(l)})$ (Rastegari et al., 2016). Similar to RTN, we also use group-wise quantization, where the scaling factor is computed separately within each group of weights.

### 3.3 Optimizing the Sub-LoRAs by Straight-Through Gradient Descent

The quantization error can be further reduced by an optimization process. This is typically accomplished by searching for an optimal reparameterization that minimizes the error after dequantization (Nagel et al., 2020). In our case, we search for $\mathbf{B}^*\mathbf{A}^*$ for the original un-quantized LoRA $\mathbf{BA}$ such that the quantization error is minimized. In particular, we perform optimization on each column of $\mathbf{B}^{(\cdot)}$ and its corresponding row of $\mathbf{A}^{(\cdot)}$, one pair at a time. This is because we have performed SVD and do not want to mix the SVD dimensions during joint optimization.[3]

Let $\boldsymbol{b}_i \in \mathbb{R}^m$ be the $i$th column of $\mathbf{B}^{(\cdot)}$, and $\boldsymbol{a}_i \in \mathbb{R}^n$ be the $i$th row of $\mathbf{A}^{(\cdot)}$; both $\boldsymbol{b}_i$ and $\boldsymbol{a}_i$ are column vectors. The goal is to find $\boldsymbol{b}_i^* \in \mathbb{R}^m$ and $\boldsymbol{a}_i^* \in \mathbb{R}^n$ to

$$\underset{\boldsymbol{b}_i^*,\boldsymbol{a}_i^*}{\text{minimize}} \ \left\| \boldsymbol{b}_i \boldsymbol{a}_i^\top - D(Q(\boldsymbol{b}_i^*))D(Q(\boldsymbol{a}_i^{*\top})) \right\|_F \tag{11}$$

where $D$ and $Q$ are one of the quantization methods in §3.2, depending on which sub-LoRA we are handling.

In other words, we reparameterize each SVD dimension such that its quantization has a smaller error. The vector $\boldsymbol{b}_i^*$ and $\boldsymbol{a}_i^*$ are initialized with $\boldsymbol{b}_i$ and $\boldsymbol{a}_i$, respectively, and are fine-tuned through gradient-based optimization. Due to the non-differentiable dequantization function, we employ the Straight-Through Estimator (STE; Bengio et al., 2013) during backpropagation, which approximates the gradient by treating the rounding function as an identity function. This allows the gradient to flow despite the non-differentiable nature of quantization.

In practice, the optimization converges within one hundred gradient steps, and thus is computationally efficient. After optimizing $\boldsymbol{b}_i^*$ and $\boldsymbol{a}_i^*$ for different $i$ values, they are put back into the two matrices $\mathbf{B}^{(\cdot)}$ and $\mathbf{A}^{(\cdot)}$ for quantization according to §3.2.

**Putting everything together**. We provide an overview of LoraQuant in Algs. 1 and 2. Alg. 1 describes the main pipeline: it first splits a LoRA into two sub-LoRAs using SVD, as detailed in §3.1, and then applies mixed-precision quantization. Specifically, the less important sub-LoRA is quantized using binary quantization, while the more important sub-LoRA is quantized using RTN, following the procedure in §3.2.

Alg. 2 presents the optimization procedure introduced in §3.3. In this step, the quantization operation is treated as a straight-through function, allowing gradients to flow through the quantizer and enabling backpropagation-based optimization of the sub-LoRAs prior to quantization.

Regarding time efficiency, the main computational bottleneck of our method is computing the SVD of the LoRA adapters. For 7B models, this takes approximately 30 minutes. However, this cost is incurred only once per customized LLM, and thus the overall overhead is small.

## 4 Experiments

We describe the experimental setups and training configurations for LoRA adapters in §4.1, including datasets, models, and evaluation protocols. We then present our main experimental results in §4.2, with detailed comparisons to baseline methods to highlight the effectiveness of our approach. We provide further analyses and ablation studies in §4.3 to better understand the impact of different design choices.

### 4.1 Settings

To evaluate the effectiveness of LoraQuant, we train LoRA for three widely used open-weight language models: LLaMA 2-7B, LLaMA 2-13B (Touvron et al., 2023), and Mistral 7B (Jiang et al., 2023). We apply the LoRAs to three distinct tasks: mathematical reasoning, code generation, and summarization. For each task, we use standard benchmark datasets for evaluation. For the mathematical reasoning task, we assess performance on the GSM8K (Cobbe et al., 2021) and MATH (Hendrycks et al., 2021) datasets using the LM Evaluation Harness framework (Gao et al., 2024); we report pass@1 accuracy as the evaluation

---

[3]In our pilot study, we also experimented with joint optimization and observed little difference. Nevertheless, we adopt this approach because it is more intuitive.

---

**Algorithm 1** LORAQUANT ($\mathbf{B}$, $\mathbf{A}$, $h$, $bits_{\text{high}}$, $bits_{\text{low}}$, $T$, $\eta$)

---

**Require:** LoRA matrices $\mathbf{B} \in \mathbb{R}^{m \times r}$ and $\mathbf{A} \in \mathbb{R}^{r \times n}$, ratio $\rho$, bitwidths $bits_{\text{high}}$ and $bits_{\text{low}}$,
   optimization steps $T$, learning rate $\eta$
 1: Compute SVD: $\mathbf{U}, \mathbf{S}, \mathbf{V}^\top \leftarrow \text{SVD}(\mathbf{BA})$
 2: $\mathbf{B}' \leftarrow \mathbf{U}\sqrt{\mathbf{S}}$   #Square root is applied element-wise
 3: $\mathbf{A}' \leftarrow \sqrt{\mathbf{S}}\mathbf{V}^\top$
 4: Find the smallest $h$ such that $\frac{\sum_{i=1}^{h} s_i^2}{\sum_{i=1}^{r} s_i^2} \geq \rho$
 5: $\mathbf{B}^{(\text{h})} \leftarrow$ first $h$ columns of $\mathbf{B}'$
 6: $\mathbf{A}^{(\text{h})} \leftarrow$ first $h$ rows of $\mathbf{A}'$
 7: $\mathbf{B}^{(\text{l})} \leftarrow$ last $r{-}h$ columns of $\mathbf{B}'$
 8: $\mathbf{A}^{(\text{l})} \leftarrow$ last $r{-}h$ rows of $\mathbf{A}'$
 9: **for** $i = 0$ **to** $h - 1$ **do**
10:    $\mathbf{B}^{(\text{h})}_{[:,i]}, \mathbf{A}^{(\text{h})}_{[i,:]} \leftarrow \text{opt}(\mathbf{B}^{(\text{h})}_{[:,i]}, \mathbf{A}^{(\text{h})}_{[i,:]}, bits_{\text{high}}, T, \eta)$
11: **end for**
12: **for** $i = 0$ **to** $r - h - 1$ **do**
13:    $\mathbf{B}^{(\text{l})}_{[:,i]}, \mathbf{A}^{(\text{l})}_{[i,:]} \leftarrow \text{opt}(\mathbf{B}^{(\text{l})}_{[:,i]}, \mathbf{A}^{(\text{l})}_{[i,:]}, bits_{\text{low}}, T, \eta)$
14: **end for**
15: $\mathbf{B}^{(\text{h})} \leftarrow \text{quantize}(\mathbf{B}^{(\text{h})}, bits_{\text{high}})$, $\mathbf{A}^{(\text{h})} \leftarrow \text{quantize}(\mathbf{A}^{(\text{h})}, bits_{\text{high}})$
16: $\mathbf{B}^{(\text{l})} \leftarrow \text{quantize}(\mathbf{B}^{(\text{l})}, bits_{\text{low}})$, $\mathbf{A}^{(\text{l})} \leftarrow \text{quantize}(\mathbf{A}^{(\text{l})}, bits_{\text{low}})$
17: **return** $(\mathbf{B}^{(\text{h})}, \mathbf{A}^{(\text{h})}), (\mathbf{B}^{(\text{l})}, \mathbf{A}^{(\text{l})})$

---

**Algorithm 2** OPT($\mathbf{B}$, $\mathbf{A}$, $bits$, $T$, $\eta$)

---

**Require:** Factor matrices $\mathbf{B}$ and $\mathbf{A}$, target bitwidth $bits$, optimization step $T$, learning rate $\eta$
 1: $\mathbf{B}_{\text{opt}} \leftarrow \mathbf{B}$, $\mathbf{A}_{\text{opt}} \leftarrow \mathbf{A}$
 2: **for** $t = 1$ **to** $T$ **do**
 3:    $\mathbf{Q}_B \leftarrow \text{quantize}(\mathbf{B}_{\text{opt}}, bits)$, $\mathbf{Q}_A \leftarrow \text{quantize}(\mathbf{A}_{\text{opt}}, bits)$
 4:    $\mathbf{B}_{\text{rec}} \leftarrow \text{dequantize}(\mathbf{Q}_B)$, $\mathbf{A}_{\text{rec}} \leftarrow \text{dequantize}(\mathbf{Q}_A)$
 5:    $\mathcal{L} \leftarrow \|\mathbf{BA} - \mathbf{B}_{\text{rec}}\mathbf{A}_{\text{rec}}\|_F$
 6:    Backpropagate $\mathcal{L}$ by straight-through estimation
 7:    $\mathbf{B}_{\text{opt}} \leftarrow \mathbf{B}_{\text{opt}} - \eta \cdot \nabla_{\mathbf{B}_{\text{opt}}}\mathcal{L}$
 8:    $\mathbf{A}_{\text{opt}} \leftarrow \mathbf{A}_{\text{opt}} - \eta \cdot \nabla_{\mathbf{A}_{\text{opt}}}\mathcal{L}$
 9: **end for**
10: **return** $\mathbf{B}_{\text{opt}}, \mathbf{A}_{\text{opt}}$

---

metric. For code generation, we evaluate our model on the HumanEval dataset (Chen et al., 2021) with the Code Generation LM Evaluation Harness framework (Ben Allal et al., 2022); the metric is the accuracy of generated programs, where a program is considered accurate if it passes all test cases. For summarization, we evaluate on the XSum dataset (Narayan et al., 2018) and report ROUGE-L (Lin, 2004) as the metric.

For LoRA adapters, we train them separately for each task, mimicking the scenario of LLM customization. For mathematical reasoning, we use the MetaMathQA dataset (Yu et al., 2023) for training, noticing that the training sets of GSM8K and MATH are too small. For code generation, we train the LoRA on the Magicoder-Eval-100-Instruct dataset (Wei et al., 2024), since HumanEval is a test-only dataset. Such training and evaluation follow the common practice in prior work (Biderman et al., 2024; Meng et al., 2024). For summarization, we use the standard training split of the XSum dataset (Narayan et al., 2018).

In all experiments, we adopt a widely used LoRA setup (Biderman et al., 2024), where the rank is set to 16 and LoRA modules are inserted into every linear layer of the transformer.[4] The training of LoRA follows that in the QLoRA study (Dettmers et al., 2023), where the base language model is quantized and frozen whereas

---

[4]We analyze how different ranks affect our method in Appendix D.

| Model | # | Method | Task | | | | Avg Perf. | Avg Bit |
|-------|---|--------|------|------|-----------|------|-----------|---------|
| | | | GSM8K | MATH | HumanEval | XSum | | |
| LLaMA 2-7B | 1 | **FP16** | 58.53 | 18.03 | 34.76 | 33.53 | 36.21 | 16 |
| | 2 | BIN | 28.89 | 6.74 | 20.12 | 24.07 | 19.95 | 1.13 |
| | 3 | RTN (1 bit) | 0.00 | 2.95 | 9.76 | 8.27 | 5.24 | 1.13 |
| | 4 | JD-Diagonal | 38.29 | 6.91 | 15.85 | 30.23 | 22.82 | 5.33 |
| | 5 | RTN (2 bits) | 49.36 | 9.94 | 29.27 | 33.23 | 30.45 | 2.14 |
| | 6 | GPTQ (2 bits) | 52.16 | 13.23 | 29.88 | 33.02 | 32.07 | 2.14 |
| | 7 | PBLLM | 50.57 | 11.20 | 28.05 | 32.42 | 30.56 | 2.83 |
| | 8 | BiLLM | 53.90 | 13.90 | 29.88 | 32.86 | 32.63 | 2.24 |
| | 9 | LoraQuant (2@0.8) | 51.25 | 10.11 | 24.39 | 32.43 | 29.55 | **1.65** |
| | 10 | LoraQuant (2@0.9) | 52.16 | 12.72 | 29.27 | 32.43 | 31.65 | 1.81 |
| | 11 | LoraQuant (3@0.8) | 53.60 | 14.57 | 29.88 | 33.35 | 32.86 | 2.16 |
| | 12 | LoraQuant (3@0.9) | **56.86** | **16.01** | **31.71** | **33.51** | **34.52** | 2.50 |
| Mistral 7B | 1 | **FP16** | 58.83 | 19.46 | 45.12 | 31.96 | 38.77 | 16 |
| | 2 | BIN | 29.26 | 9.18 | 18.90 | 8.74 | 16.52 | 1.13 |
| | 3 | RTN (1 bit) | 13.42 | 11.37 | 7.32 | 15.43 | 11.88 | 1.13 |
| | 4 | JD-Diagonal | 0.00 | 6.49 | 3.66 | 15.08 | 6.31 | 5.33 |
| | 5 | RTN (2 bits) | 26.08 | 5.31 | 34.15 | 31.42 | 24.24 | 2.14 |
| | 6 | GPTQ (2 bits) | 40.26 | 9.69 | 31.71 | 32.46 | 28.53 | 2.14 |
| | 7 | PBLLM | 50.04 | 17.94 | 38.41 | 33.43 | 34.96 | 2.83 |
| | 8 | BiLLM | 50.95 | 14.41 | 42.07 | 32.32 | 34.94 | 2.24 |
| | 9 | LoraQuant (2@0.8) | 52.08 | 16.43 | 35.98 | 33.26 | 34.44 | **1.85** |
| | 10 | LoraQuant (2@0.9) | **53.90** | 17.94 | 39.63 | **33.48** | 36.24 | 1.97 |
| | 11 | LoraQuant (3@0.8) | 51.18 | 18.45 | **44.51** | 33.01 | 36.79 | 2.56 |
| | 12 | LoraQuant (3@0.9) | 53.75 | **18.70** | 43.90 | 32.75 | **37.28** | 2.80 |
| LLaMA 2-13B | 1 | **FP16** | 61.79 | 18.79 | 46.34 | 35.16 | 40.52 | 16 |
| | 2 | BIN | 27.37 | 9.60 | 21.34 | 31.97 | 22.57 | 1.13 |
| | 3 | RTN (1 bit) | 0.00 | 4.13 | 13.41 | 1.35 | 4.72 | 1.13 |
| | 4 | JD-Diagonal | 46.17 | 8.51 | 18.91 | 33.14 | 26.68 | 5.33 |
| | 5 | RTN (2 bits) | 53.30 | 14.49 | 32.93 | 34.54 | 33.81 | 2.14 |
| | 6 | GPTQ (2 bits) | 57.77 | 15.08 | 36.59 | 34.56 | 36.00 | 2.14 |
| | 7 | PBLLM | 59.06 | 16.34 | 32.93 | 34.08 | 35.60 | 2.83 |
| | 8 | BiLLM | 59.89 | 17.02 | 36.59 | 34.79 | 37.07 | 2.24 |
| | 9 | LoraQuant (2@0.8) | 56.63 | 15.08 | 30.49 | 34.05 | 34.06 | **1.65** |
| | 10 | LoraQuant (2@0.9) | 57.39 | 15.67 | 35.98 | 34.27 | 35.83 | 1.81 |
| | 11 | LoraQuant (3@0.8) | 60.05 | **18.03** | 35.37 | 34.74 | 37.05 | 2.17 |
| | 12 | LoraQuant (3@0.9) | **60.80** | 17.44 | **39.02** | **34.95** | **38.06** | 2.50 |

Table 1: Performance and average bitwidth for different methods. "Avg Perf." refers to average performance. In Rows 5–12, we **bold** the best task performance, as well as the least average bit among models.

the LoRA is trained in half precision (FP16).[5] Training hyperparameters are provided in Appendix A. After training, we apply our LoraQuant method and other quantization baselines to the LoRA weights and evaluate the performance of each approach.

## 4.2 Main Results

We present the main results in Tab. 1, where we consider the following popular baselines for quantizing LoRA. **GPTQ** (Frantar et al., 2023) quantizes weights sequentially and adjusts the remaining weights to reduce quantization error. **RTN** and **BIN** are the standard methods introduced in §3.2. **PB-LLM** (Shang et al., 2024) performs mixed-precision quantization with some weight being binarized and others maintained in full precision; note that an additional indicator bit is needed for each weight. **BiLLM** (Huang et al., 2024) works similarly, except that each column must be either quantized or maintained in full precision while

---

[5]In a pilot experiment, we experimented with the full-precision LLaMA 2-7B model on the GSM8K dataset. We observe minimal performance degradation by using QLoRA for the base model: 58.53% accuracy with QLoRA and 59.28% with full precision. Thus, we used QLoRA to quantize the base model for efficiency considerations.

using a split binarization strategy. All baselines perform group quantization with a group size of 128, which is a common practice in the literature (Lin et al., 2024; Frantar et al., 2023).

We also consider **JD-Diagonal** (Gabrielsson et al., 2024), which is not a quantization method but is related to the general objective of our research: reducing memory for multiple LoRAs. Specifically, JD-Diagonal clusters different LoRAs and performs weight sharing, while introducing $r$-many additional parameters for each task, where $r$ is the adapter's rank. In our evaluation, we treat the three tasks as a cluster for weight sharing.

Our comparison focuses on two aspects: (1) output quality measured by the standard metric of each task, and (2) average number of bits per LoRA parameter,[6] which reflects the effectiveness of memory saving. When computing the average bits, we also consider the scale and the zero-point parameters in our computation. The values reported in Tab. 1 represent the average bitwidth across the three task-specific adapters. For detailed per-adapter bitwidth, refer to Appendix B.

As shown in Tab. 1, FP16 (Row 1) achieves the highest overall performance across all models, which is expected. Various quantization methods (Rows 2–3 and Rows 5–8) provide a spectrum of performance–memory tradeoff. However, all previous methods require at least two bits to achieve reasonable performance. As we see, binary quantization (Row 2) yields 30-point degradation in accuracy on GSM8K with LLaMA 2-7B, whereas RTN (1 bit) almost collapses the model, leading to extremely poor accuracy.

We also observe that JD-Diagonal fails to achieve reasonable performance in our experiments. We hypothesize that the discrepancy between our experiment and that in Gabrielsson et al. (2024) is due to the tasks and evaluation metrics. Gabrielsson et al. (2024) only consider simple tasks with in-context learning samples, which can already be handled well by the base model. In our pilot study, we find that even reducing the LoRA rank to 1 via SVD has little impact on the performance of their trained adapters, suggesting that the LoRA parameters contribute minimally in their settings. Also, they only use ROUGE-L as their metric, while in our setting math and coding require exact matching. Appendix F provides additional analysis on JD-Diagonal.

We then examine our LORAQUANT approach (Rows 9–12). LORAQUANT performs mixed-precision quantization, where the more important sub-LoRA is quantized to $i$ bits ($i = 2$ or $3$), and the less important sub-LoRA is always quantized to 1 bit. The fraction is controlled by the hyperparameter $\rho$, which is the total variation explained (§3.1). Our variant is denoted by LORAQUANT($i@\rho$).

As seen, the 2@0.8 and 2@0.9 variants (Rows 9–10) consistently operate under 2 bits, while achieving high performance comparable to, or even higher than, PB-LLM and BiLLM, which are also mixed-precision quantization but use more bits than ours. To enable a fair comparison with these binarization methods at a similar average bitwidth, we also experiment with quantizing the more important sub-LoRA to 3 bits. Our 3@0.8 and 3@0.9 variants (Rows 11–12) consistently achieve higher task performance than both PB-LLM and BiLLM.

To conclude, our LORAQUANT method is able to reduce the memory of LoRAs by a large margin while achieving comparable performance with full precision baseline. Our approach also largely advances ultra-low-bit quantization with less than two bits per parameter in the LoRA setting.

### 4.3 In-Depth Analyses

We conduct in-depth analyses with the LLaMA-2-7B model on the GSM8K and MATH datasets. We restrict our analyses to this setting due to computational constraints.

**Sub-LoRA split strategy.** In Fig. 2, we compare splitting sub-LoRAs using SVD against two baseline strategies: (i) selecting columns of $\mathbf{B}$ and the corresponding rows of $\mathbf{A}$ at random, and (ii) selecting the high-precision components according to the magnitude of $\mathbf{b}_i\mathbf{a}_i$ (where $\mathbf{b}_i$ denotes the $i$th column of $\mathbf{B}$ and $\mathbf{a}_i$ denotes the $i$th row of $\mathbf{A}$), measured by its Frobenius norm. The intuition behind the norm-based strategy

---

[6]Notice that our paper focuses on LLM customization where massive numbers of LoRAs are loaded to a base, high-performing LLM. Therefore, our base LLM follows a standard QLoRA treatment and its parameter size (which is a constant) is not considered by the metric in the main text. Appendix C provides a memory analysis with the base LLM.

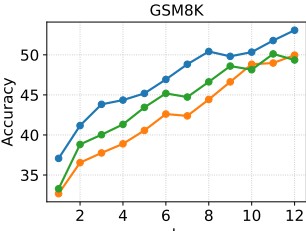 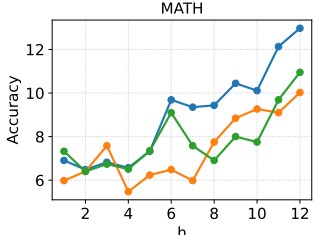 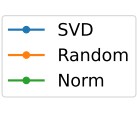

Figure 2: Comparison of sub-LoRA splitting strategies. Here, $h$ denotes the rank of the high-precision sub-LoRA and is fixed globally for all LoRAs at different layers in a model.

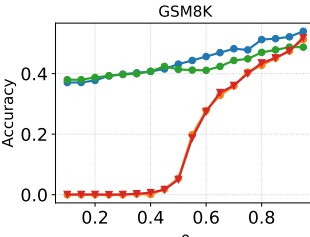 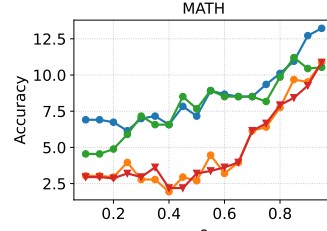 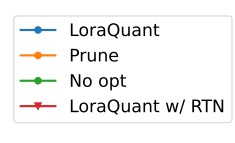

Figure 3: Study on optimization and quantization of LORAQUANT. **LoraQuant** is the proposed method. **Prune** truncates the less important sub-LoRA components. **No Opt** omits the gradient-based optimization step. **LoraQuant w/ RTN** replaces the specialized binarization with 1-bit RTN quantization.

is that components with larger norms contribute more substantially to the overall LoRA update, and thus are more suitable for higher-precision quantization. In this analysis, we do not choose $h$ dynamically (§3.1) but directly set the value of $h$ to ensure a fair comparison with the random and norm-based strategies. As seen in the plots, our SVD split strategy generally outperforms the other methods. This is consistent with our intuition that SVD identifies the important dimensions, for which we should preserve more information during quantization.

**Ablation study.** In Fig. 3, we analyze the effect of different components of our approach. First, we ablate the gradient-based optimization, which searches for reparameterization (after splitting the sub-LoRAs) to reduce quantization error (§3.3). Fig. 3 shows that the optimization generally improves the performance of LORAQUANT, with higher improvement for higher ratios; therefore, we adopt the optimization process in our approach.

Next, we ablate our approach by pruning the less important sub-LoRA entirely in order to test whether it contributes to performance. As shown in Fig. 3, pruning collapses the model at lower ratios, which is expected. As the ratio increases, the performance of pruning also increases, but is consistently lower than our LORAQUANT. This verifies that the less important sub-LoRA, even quantized to 1 bit per weight, still plays a role in the model.

We also experiment an alternative variant of LORAQUANT, where the less important sub-LoRA is quantized by 1-bit RTN, instead of sign-based binarization (§3.2). This setting performs similarly to pruning the less important sub-LoRA, as 1-bit RTN effectively maps lots of values to zero. The result justifies our different quantization methods used: RTN for the important sub-LoRA and sign-based binary quantization for the less important one.

In Fig. 4, we compare our ratio-based dynamic $h$ selection strategy with a static approach that fixes $h$ to a single global value across all adapters. The results show that dynamically selecting $h$ generally leads to better performance, particularly when a slightly higher bit budget is allowed (e.g., more than 1.5 bits), which corresponds to a more practical deployment setting. This improvement suggests that different adapters benefit from different splits depending on their spectral characteristics. Nevertheless, the static variant of

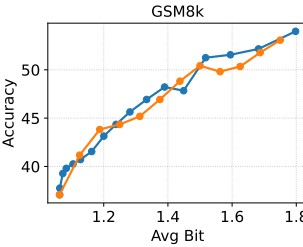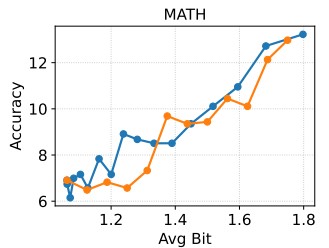

Figure 4: Comparison of $h$ selection strategy. **Ratio** denotes our method explained in §3.1, where the ratio hyperparameter varies from 0.1 to 0.95 in increments of 0.05, while **Static** sets $h$ to a fixed value ranging from 1 to 12.

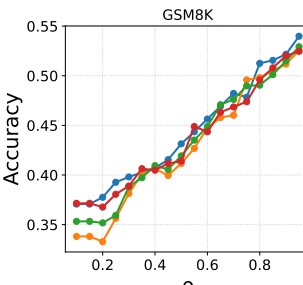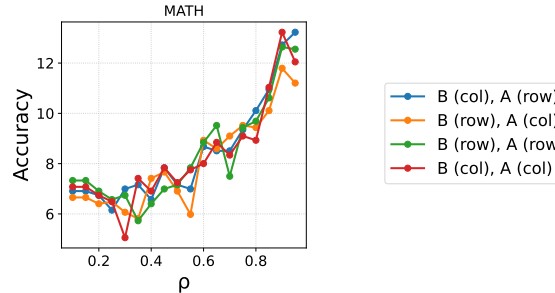

Figure 5: Study on the column-wise and row-wise quantization of LoRaQuant. Each model variant is denoted as **B (__), A (__)**, where each underscore can be either **col** or **row**. Here, **col** indicates column-wise quantization and **row** indicates row-wise quantization of the corresponding component.

our method still performs comparably, indicating that our approach remains robust even when a fixed $h$ is used.

**Quantizing along Column or Row**. In our approach, we adopt group quantization, where $\mathbf{B}'$ is quantized in a column-wise manner and $\mathbf{A}'$ is quantized row-wise. This follows naturally from the SVD reparameterization of the adapter weight: the square root of each singular value is multiplied with the corresponding columns of $\mathbf{U}$ to form $\mathbf{B}'$, and with the corresponding rows of $\mathbf{V}^\top$ to form $\mathbf{A}'$. Under this scheme, the singular values can be cleanly absorbed into the RTN scaling factors stored in FP16, which ensures that the magnitude of each singular component is preserved exactly and does not introduce additional quantization error.

We also explore alternative quantization strategies. Our hypothesis is that column-wise quantization of $\mathbf{B}'$ and row-wise quantization of $\mathbf{A}'$ should yield stronger performance. The results of this comparison are presented in Fig. 5. As shown, this hypothesis holds for the GSM8K dataset, where this configuration performs best, but on the MATH dataset, no single strategy wins consistently. That being said, the performance difference remains small, and we adopt column-wise quantization of $\mathbf{B}'$ and row-wise quantization of $\mathbf{A}'$ as the default setting in our approach since it makes more sense intuitively.

## 5 Conclusion

In this paper, we address the problem of memory reduction when multiple LoRAs are loaded simultaneously in the scenario of LLM customization (e.g., for different tasks and/or users). We propose LoRaQuant, a mixed-precision quantization method tailored for LoRAs, where we split each LoRA into two sub-LoRAs via SVD. The more important sub-LoRA is preserved with more bitwidth, whereas the less important sub-LoRA is quantized to one bit; we further adopt straight-through gradient optimization to improve the quantization. Experimental results demonstrate that LoRaQuant maintains strong performance even under extremely

low bitwidth settings. We further present in-depth analyses to verify the effectiveness of each component in our approach.

**Limitation and future work.** In the experiments, we have tried four datasets and three language models (12 evaluations in total). Due to the limit of our computing resources, we are unable to perform commercial scale experiments (e.g., having millions of LoRAs). Luckily, our method treats different LoRAs independently, and thus is easily scalable, as opposed to Gabrielsson et al. (2024) who require recomputing the parameters of each task cluster. Appendix C analyzes the memory saving of LoRaQuant when loading multiple adapters. Results confirm the practical values of our approach if the number of customized LLMs scales, and we leave the commercial applications of our method to the industry.

Moreover, our SVD-based mixed-precision quantization is specific to LoRA models and is not directly applicable to a full weight matrix. This is because, if a matrix is not low-rank, splitting a matrix into two products of sub-matrices would in fact increase the number of parameters. In the future work, we plan to adapt our method to general weight matrices by truncating the SVD dimensions, which is beyond the scope of this paper. That being said, our LoRaQuant approach can be combined with other quantization methods, and in our experiments, we have already used a four-bit quantized base models. Another promising direction is to extend our method to support multiple sub-LoRAs of different bitwidths instead of just two, which could yield a better balance of performance and model compression.

## 6 Acknowledgments

We thank all reviewers and editors for their insightful feedback and constructive comments. This research was supported in part by the Natural Sciences and Engineering Research Council of Canada (NSERC), a Mitacs Accelerate project, the Amii Fellow Program, the Canada CIFAR AI Chair Program, and the Digital Research Alliance of Canada.

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

## A   Implementation Details

We follow the hyperparameter settings of Biderman et al. (2024), except that we reduce the batch size because they train across multiple GPUs while we only use a single GPU. Below are key hyperparameters:

- **optimizer:** adamw_torch ($\beta = [0.9, 0.95]$)

| Model | Method | GSM8k & MATH | HumanEval | Xsum |
|---|---|---|---|---|
| LLaMA 2-7B | LORAQUANT (2@0.8) | 1.65 | 1.55 | 1.74 |
| | LORAQUANT (2@0.9) | 1.82 | 1.74 | 1.89 |
| | LORAQUANT (3@0.8) | 2.17 | 1.98 | 2.34 |
| | LORAQUANT (3@0.9) | 2.51 | 2.34 | 2.65 |
| Mistral 7B | LORAQUANT (2@0.8) | 1.86 | 1.82 | 1.85 |
| | LORAQUANT (2@0.9) | 1.98 | 1.95 | 1.97 |
| | LORAQUANT (3@0.8) | 2.59 | 2.51 | 2.58 |
| | LORAQUANT (3@0.9) | 2.83 | 2.76 | 2.81 |
| LLaMA 2-13B | LORAQUANT (2@0.8) | 1.61 | 1.56 | 1.77 |
| | LORAQUANT (2@0.9) | 1.79 | 1.75 | 1.91 |
| | LORAQUANT (3@0.8) | 2.09 | 2.00 | 2.40 |
| | LORAQUANT (3@0.9) | 2.44 | 2.37 | 2.69 |

Table 2: Average bitwidth of LORAQUANT variants.

- **learning rate:** $2 \times 10^{-4}$

- **scheduler:** cosine_with_warmup ($\alpha_f = 0.01$, $t_{\text{warmup}} = 0.3 \, \text{dur}$)

- **weight decay:** 0

- **precision:** fp16

- **device_train_microbatch_size:** 6

- **gradient clipping:** norm (threshold $= 1$)

- **num_epochs:** 2

- **num_gpus:** 1

For both mathematical reasoning and summarization tasks, we set **max_seq_len** to 1024. For the code domain, we use **max_seq_len** = 4096. The **batch_size** is 16 for Mistral and LLaMA-2-7B, and 8 for LLaMA-2-13B.

## B    Average Bitwidth of LoraQuant

In our work, we use AvgBits to measure the memory usage, given by

$$\text{AvgBits} = \frac{\text{total bits for LoRAs across different layers}}{\text{total \# of LoRA parameters across different layers}} . \tag{12}$$

In the main experiment (Tab. 1), we present the average bitwidth across multiple tasks, due to table formatting reasons. In this appendix, we show the bitwidth for individual tasks in Tab. 2. Notice that we have a dynamic allocation strategy, so the average bits vary slightly based on the task.

## C    Memory Analysis for LLM Customization

In this appendix, we provide an analysis of memory savings when loading different numbers of adapters, where we use the average bitwidth reported in the main experiments (specifically, the 2@0.8 setup on the GSM8K dataset in Tab. 1). As shown in Fig. 6, the memory usage of FP16 grows substantially when the number of LoRAs increases. For instance, loading only 50 LoRAs requires 2 times more memory than the base LLM. This justifies our motivation that the memory of multiple LoRAs can build up and become non-negligible, even if each LoRA is relatively lightweight. On the contrary, the memory grows slowly in our LORAQUANT method, showing its practical values for LLM customization.

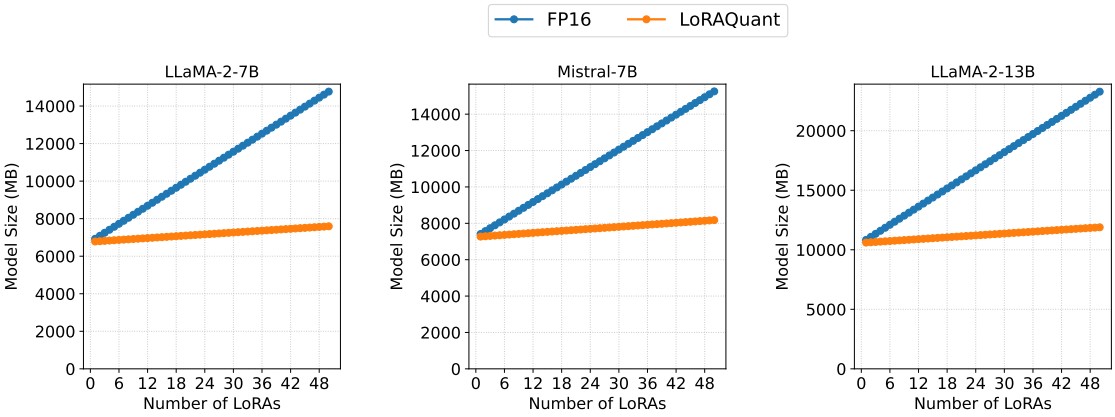

Figure 6: Memory usage when loading multiple LoRAs and the base LLM.

## D  Training Adapters with Different Ranks

In our main experiments, we train adapters using LoRA with rank 16. However, to demonstrate that our method is not dependent on the LoRA rank, we evaluate ranks $r = 8$ and $r = 32$ in this appendix. We restrict this analysis to Llama2-7B on the math datasets due to time constraint. The results are summarized in Tab. 3. As shown, our method preserves the same overall trends observed in Tab. 1, indicating that its effectiveness is consistent across different LoRA ranks.

## E  SVDQuant Analysis

In this section, we analyze SVDQuant (Li et al., 2025), a quantization method based on singular value decomposition (SVD). In SVDQuant, a weight matrix $\mathbf{W} \in \mathbb{R}^{m \times n}$ is decomposed as

$$\mathbf{W} = \mathbf{W}' + \mathbf{B}_W \mathbf{A}_W,$$

where $\mathbf{B}_W \in \mathbb{R}^{m \times r}$ and $\mathbf{A}_W \in \mathbb{R}^{r \times n}$ correspond to the top-$r$ components of the SVD of $\mathbf{W}$. These components are stored in higher precision (full precision in their work), while the residual matrix $\mathbf{W}'$ is quantized to a lower precision.

In contrast, our work focuses on quantizing LoRA adapters rather than full weight matrices. However, we still compare our method with SVDQuant by individually applying the SVDQuant procedure to matrices $\mathbf{B}$ and $\mathbf{A}$ in a LoRA adapter. As shown in Tab. 4, SVDQuant generally underperforms our method in both performance and average bit count.

We hypothesize that this performance gap arises because SVDQuant focuses on the individual factors $\mathbf{B}$ and $\mathbf{A}$, while in LoRA-based methods the effective contribution to the model comes from their product $\mathbf{BA}$. By directly modeling and quantizing this product, our method better preserves the information that is most relevant to downstream performance.

## F  JD-Diagonal Analysis

In this section, we examine the JD-Diagonal method (Gabrielsson et al., 2024) in greater detail and analyze why it does not perform well in our setting. JD-Diagonal clusters multiple LoRA adapters and jointly compresses them under the assumption that the adapters share a high degree of structural similarity. In the original work, the authors demonstrate successful clustering of up to 100 LoRA adapters with minimal performance degradation. However, in our experiments, the method exhibits severe degradation even when clustering as few as three LoRA adapters.

| Model | # | Method | Tasks | | Avg Perf. | Avg Bit |
|-------|---|--------|-------|-------|-----------|---------|
| | | | GSM8K | MATH | | |
| LoRA (r=32) | 1 | **FP16** | 58.07 | 20.72 | 39.39 | 16 |
| | 2 | BIN | 15.39 | 6.82 | 11.11 | 1.13 |
| | 3 | RTN (1 bit) | 0.00 | 3.62 | 1.81 | 1.13 |
| | 4 | RTN (2 bits) | 51.48 | 9.94 | 30.71 | 2.14 |
| | 5 | GPTQ (2 bits) | 52.62 | 15.16 | 33.89 | 2.14 |
| | 6 | PBLLM | 49.36 | 12.64 | 31.00 | 2.83 |
| | 7 | BiLLM | 53.15 | 16.68 | 34.92 | 2.24 |
| | 8 | LORAQUANT (2@0.8) | 50.11 | 11.37 | 30.74 | **1.51** |
| | 9 | LORAQUANT (2@0.9) | 50.80 | 13.56 | 32.18 | 1.67 |
| | 10 | LORAQUANT (3@0.8) | 55.72 | 17.02 | 36.37 | 2.02 |
| | 11 | LORAQUANT (3@0.9) | **56.41** | **17.52** | **36.96** | 2.35 |
| LoRA (r=16) | 1 | **FP16** | 58.53 | 18.03 | 38.28 | 16 |
| | 2 | BIN | 28.89 | 6.74 | 17.82 | 1.13 |
| | 3 | RTN (1 bit) | 0.00 | 2.95 | 1.48 | 1.13 |
| | 4 | RTN (2 bits) | 49.36 | 9.94 | 29.65 | 2.14 |
| | 5 | GPTQ (2 bits) | 52.16 | 13.23 | 32.70 | 2.14 |
| | 6 | PBLLM | 50.57 | 11.20 | 30.88 | 2.83 |
| | 7 | BiLLM | 53.90 | 13.90 | 33.90 | 2.24 |
| | 8 | LORAQUANT (2@0.8) | 51.25 | 10.11 | 30.68 | **1.65** |
| | 9 | LORAQUANT (2@0.9) | 52.16 | 12.72 | 32.44 | 1.82 |
| | 10 | LORAQUANT (3@0.8) | 53.60 | 14.57 | 34.09 | 2.17 |
| | 11 | LORAQUANT (3@0.9) | **56.86** | **16.01** | **36.44** | 2.51 |
| LoRA (r=8) | 1 | **FP16** | 54.36 | 16.68 | 35.52 | 16 |
| | 2 | BIN | 40.19 | 5.31 | 22.75 | 1.13 |
| | 3 | RTN (1 bit) | 0.00 | 4.63 | 2.32 | 1.13 |
| | 4 | RTN (2 bits) | 49.97 | 12.47 | 31.22 | 2.14 |
| | 5 | GPTQ (2 bits) | 47.23 | 11.29 | 29.26 | 2.14 |
| | 6 | PBLLM | 49.73 | 11.04 | 30.38 | 2.83 |
| | 7 | BiLLM | 52.99 | 12.30 | 32.65 | 2.24 |
| | 8 | LORAQUANT (2@0.8) | 44.20 | 7.75 | 25.98 | **1.50** |
| | 9 | LORAQUANT (2@0.9) | 46.93 | 10.70 | 28.81 | 1.66 |
| | 10 | LORAQUANT (3@0.8) | 50.64 | 12.47 | 31.56 | 2.00 |
| | 11 | LORAQUANT (3@0.9) | **52.84** | **13.14** | **32.99** | 2.33 |

Table 3: Performance and average bitwidth for different rank configuration. "Avg Perf." refers to average performance. In Rows 4–11, we **bold** the best task performance, as well as the least average bit among models.

In their work, the authors train thousands of task-specific adapters for Mistral-7B and demonstrate strong performance when clustering them jointly. To better understand this behavior, we analyze the properties of the adapters trained in their setting. In Fig. 7, we plot the average explained variance ratio for each rank of the adapters, computed over all adapters corresponding to three representative tasks and then averaged within each task.

The explained variance ratio reveals a strong concentration of information in the leading ranks of the adapters trained in their setting. As shown in Fig. 7, for all three tasks, the first few ranks account for a large fraction of the total variance, while subsequent ranks contribute only marginal gains. This indicates that a significant portion of each adapter is redundant and can be safely pruned or shared across tasks with minimal loss.

In contrast, when examining our three adapters trained on the same model in Fig. 7, we do not observe such dominant spikes in the leading ranks. Instead, the explained variance ratio exhibits a more linear decay, suggesting that each rank contains meaningful information. As a result, forcing these adapters into a shared low-rank structure leads to substantial information loss, which explains the poor performance of JD-Diagonal in our setting even when clustering only a small number of adapters.

| Model | Method | GSM8k | MATH | HumanEval | Xsum | Avg Perf. | Avg Bit |
|---|---|---|---|---|---|---|---|
| LLaMA 2-7B | **FP16** | 58.53 | 18.03 | 34.76 | 33.53 | 36.21 | 16 |
| | RTN (2 bits) | 49.36 | 9.94 | **29.27** | **33.23** | **30.45** | 2.14 |
| | LoraQuant (2@0.8) | **51.25** | **10.11** | 24.39 | 32.43 | 29.55 | **1.65** |
| | SVDQuant | 45.41 | 9.27 | 28.05 | 32.53 | 28.81 | 2.44 |
| Mistral 7B | **FP16** | 58.83 | 19.46 | 45.12 | 31.96 | 38.77 | 16 |
| | RTN (2 bits) | 26.08 | 5.31 | 34.15 | 31.42 | 24.24 | 2.14 |
| | LoraQuant (2@0.8) | **52.08** | **16.43** | **35.98** | **33.26** | **34.44** | **1.85** |
| | SVDQuant | 6.29 | 4.47 | 30.48 | 30.55 | 17.95 | 2.62 |
| LLaMA-13B | **FP16** | 61.79 | 18.79 | 46.34 | 35.16 | 40.52 | 16 |
| | RTN (2 bits) | 53.30 | 14.49 | **32.93** | **34.54** | 33.81 | 2.14 |
| | LoraQuant (2@0.8) | **56.63** | **15.08** | 30.49 | 34.05 | **34.06** | **1.65** |
| | SVDQuant | 52.16 | 13.65 | 25.00 | 34.01 | 31.21 | 2.40 |

Table 4: Comparing SVDQuant and LoraQuant.

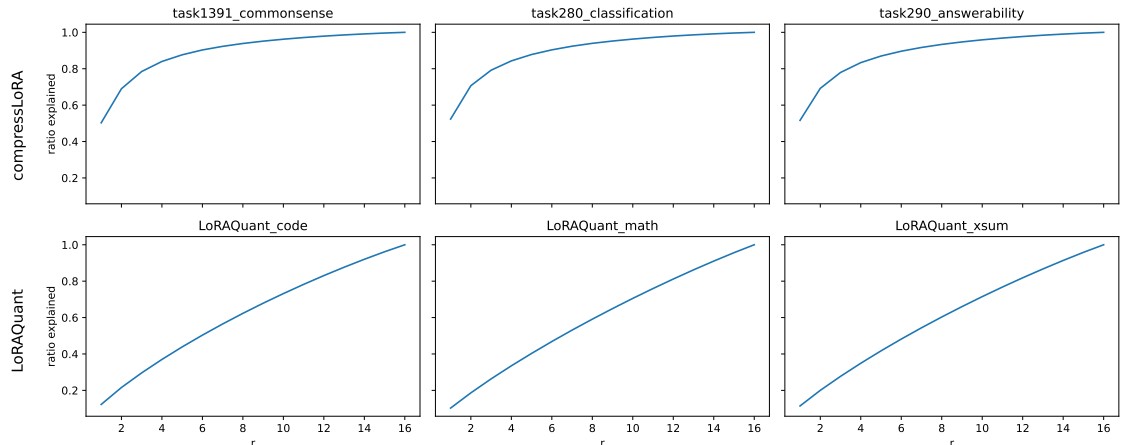

Figure 7: Explained variance ratio across ranks for six tasks. The top row shows adapters trained with CompressLoRA (Gabrielsson et al., 2024), while the bottom row corresponds to our adapters.

