# OpenReview forum: "LoraQuant: Mixed-Precision Quantization of LoRA to Ultra- Low Bits for LLM Customization"
_TMLR — Accepted by TMLR_

### Review · Reviewer_XC6t · 2026-02-26

**Summary Of Contributions:**

This paper proposes a novel method to quantize LoRA modules: using SVD to decompose LoRA matrices into two parts--the more important one and the less important one, and quantize with different precision according to their importance. It gives tailored method on how to split more/less important module and how to quantize in LoRA.

Strengths:
1. The paper is organized well, and the logic is clear. Main experiments are across three models, demonstrating the effectiveness of the proposed method. Ablational experiments have also been fully studied to reveal the role of different components.
2.  It sounds reasonable to compress all parameters to low precision rather than keeping some of them in full precision and only compress the rest, if an extremely low average precision is desired. Also, the design of assigning different precision for different importance is natural.

Weaknesses:
1. I don't really understand Appendix D. What do you mean by "SVDQuant quantizes the low-rank factors BW and AW individually, whereas our approach considers the SVD of the product BA and applies mixed-precision quantization directly to this product. " ? Please explain this clearly.
2. I don't see any point in why the process in Section 3.3(Algorithm 2) could help as the preprocessing of quantization. This optimization converges at some point with 0 quantization error, e.g with the expressiveness of 1bit/2bit. However, since the loss is independent of training data(only quantization error), this process should be similar to quantize directly.
3. In your setting, how is the proportion of saved memory? How many LoRA modules would be used in practical scenarios? This would be related to your significance.

**Audience:**

Yes

**Audience Explanation:**

I think some people might be interested in the extremely low precision quantization of LoRA modules. However, the novelty of this method is highly limited given SVDQuant.

**Claims And Evidence:**

Yes

**Claims Explanation:**

Table 1 presents comprehensive experimental evidence about the effectiveness and memory efficiency of LoRAQuant compared to various baseline methods.

**Requested Changes:**

My main concerns are listed in the weakness part of the summary and the first one is most important. Apart from that:

1. The experiments are mainly about extremely low average precision. Is there any evaluation and comparison for all methods on other intervals(e.g, ~4bits?) It's acceptable if you don't have, but I think it could strengthen the work in my view.
2. A relatively minor question: how is the time efficiency of your algorithm, especially when scaling up to multi-LoRA situation?

---

> ### Author Response · Authors · 2026-04-29
> **Response by Authors**
>
> We thank the reviewer for recognizing the paper’s clarity and the comprehensive experimental and ablation studies. Below we’ll address the weaknesses and change requests point by point.
> ### Weakness 1
> > I don't really understand Appendix D. What do you mean by "SVDQuant quantizes the low-rank factors \(BW\) and \(AW\) individually, whereas our approach considers the SVD of the product \(BA\) and applies mixed-precision quantization directly to this product." Please explain this clearly.
>
> Thanks for the question. The original SVDQuant paper focuses on quantizing entire weight matrices. In other word, a weight matrix $\mathbf{W} \in \mathbb{R}^{m × n}$ is decomposed as $\mathbf{W} = \mathbf{W}' + \mathbf{B_W A_W}$, where $\mathbf{B_W} \in \mathbb{R}^{m × r}$ and $\mathbf{A_W} \in \mathbb{R}^{r × n}$ are the top-$r$ components of the SVD of $\mathbf{W}$, stored in higher precision;the residual matrix $\mathbf{W}'$ is quantized to a lower precision.
>
> In contrast, our work focuses on quantizing LoRA adapters rather than full weight matrices. However, we still compare our method with SVDQuant by individually applying the SVDQuant procedure to matrices $\mathbf{B}$ and $\mathbf{A}$ in a LoRA adapter. We then compare the performance in Appendix D (now Appendix E). In the revision, we provided rationales for this comparison and clarified how SVDQuant was applied.
> ### Weakness 2
> >  I don't see any point in why the process in Section 3.3(Algorithm 2) could help as the preprocessing of quantization. This optimization converges at some point with 0 quantization error, e.g with the expressiveness of 1bit/2bit. However, since the loss is independent of training data(only quantization error), this process should be similar to quantize directly.
>
> This process helps reduce direct quantization error $|\mathbf{W} - D(Q(\mathbf{W}*))|$, i.e., without considering the training data. The optimization typically does not converge to 0 quantization error, as quantization reduces the number of bits and is not invertible.. However, the quantization error can be reduced to some extent [1]. It does so by applying a gradient-based optimization to find weights that are closer to the original values **after** dequantization. We also demonstrate how this improves performance in our ablation studies. In our revision, we provided further explanation at the beginning of Section 3.3.
>
> ### Weakness 3
> > In your setting, how is the proportion of saved memory? How many LoRA modules would be used in practical scenarios? This would be related to your significance.
>
> This is an insightful question! We have presented this analysis in Appendix C of the original submission, showing how much memory our method saves as we scale the number of LoRA adapters The practical scenario we consider is LoRA-based personalization, where the number of adapters may scale with the number of users of a large model. Results show that, even with 50 LoRAs, we already save almost 50% of the total parameters (considering the base LLM). In practice, LLM providers may have millions of LoRAs (although companies do not disclose their exact usages), and thus our work is of great practical significance.
> In the revision, we provide a more detailed reference to Appendix C in Section 5.
>
> ### Change 1
>
> > The experiments are mainly about extremely low average precision. Is there any evaluation and comparison for all methods on other intervals(e.g, ~4bits?) It's acceptable if you don't have, but I think it could strengthen the work in my view.
>
> Thanks for the suggestion! We do not report performance at 4-bit precision, as there is minimal degradation compared to full precision; instead, we focus on lower-precision settings.
> To show this, we compare the result of our adapter for llama2_7b and llama2_13b on the GSM8k dataset below.  As seen, there is minimal degradation in the RTN-3 and RTN-4 settings, thus not being the focus in our scenario.
> | Model      | FP    | RTN-3 | RTN-4 |
> |------------|-------|-------|-------|
> | Llama2-7B  | 58.53 | 57.85 | 58.68 |
> | Llama2-13B | 61.79 | 61.94 | 62.17 |
> ### Change 2
> >  A relatively minor question: how is the time efficiency of your algorithm, especially when scaling up to multi-LoRA situation?
>
> The main computational bottleneck of our method is computing the SVD of the LoRA adapters. For 7B models, this takes approximately 30 minutes. However, this cost is incurred only once per user. We include an additional paragraph in Section 3.3.
>
> > Additional comment on novelty in comparison with SVDQuant
>
> Regarding the concern about novelty with respect to SVDQuant, we hope that our explanation for Weakness 1 has adequately addressed the reviewer’s concern.
>
> [1] Li, Gong, et al. “BRECQ: Pushing the Limit of Post-Training Quantization by Block Reconstruction” ICLR 2021

---

> > ### Comment · Reviewer_XC6t · 2026-04-30
> >
> > Thank you for your response. My questions have been addressed, but I do insist my comment on the novelty.

---

### Review · Reviewer_EVQa · 2026-03-14

**Summary Of Contributions:**

This paper introduces LoraQuant, a post-training quantization method designed specifically for LoRA adapters. The work focuses on reducing the memory footprint of LoRA adapters in scenarios where many adapters must be stored or loaded simultaneously.

Strengths:
1. Clear motivation and simple, intuitive method design.
2. The paper is clearly written and well organized.

Weaknesses:
1. The experiments only consider LoRA with rank r=16. It would be interesting to see whether the proposed method remains effective for different LoRA ranks (e.g., r=8, r=32), especially when the singular value spectrum becomes flatter.

2. The LoRA adapters are trained following the QLoRA setup. It would be interesting to evaluate whether the proposed method generalizes to LoRA adapters trained on full-precision base models.

**Audience:**

Yes

**Audience Explanation:**

I think the findings of this paper can to be of interest to a portion of the TMLR audience, particularly practitioners working on efficient training and deployment of large language models. Parameter-efficient fine-tuning methods such as LoRA are widely used in practice, and reducing the memory footprint of adapters while maintaining model performance can be a practical problem.

**Claims And Evidence:**

Yes

**Claims Explanation:**

The claims made in the paper are generally supported by convincing empirical evidence. The authors conduct experiments across multiple  language models (LLaMA-2-7B, LLaMA-2-13B, and Mistral-7B) and evaluate on diverse downstream tasks, including mathematical reasoning (GSM8K and MATH), code generation (HumanEval), and summarization (XSum). These experiments demonstrate that the proposed LoraQuant method can achieve competitive performance while significantly reducing the average bitwidth of LoRA parameters.

**Requested Changes:**

1. How sensitive is performance to the LoRA rank (e.g., r=8, r=32)? Does the method degrade when the singular spectrum is flatter?
2. How does LoraQuant perform when applied to LoRA trained in full precision (without QLoRA base model)?

---

> ### Author Response · Authors · 2026-04-29
> **Response by Authors**
>
> Thanks for recognizing the paper’s clarity and the intuitive design of our model. We would like to address the concerns raised by the reviewer as follows.
>
> ### Weakness 1
> > The experiments only consider LoRA with rank r=16. It would be interesting to see whether the proposed method remains effective for different LoRA ranks (e.g., r=8, r=32), especially when the singular value spectrum becomes flatter.
>
> We choose $r$ = 16 as it is a commonly used setting in prior work on training LoRA adapters. We’re grateful to the reviewer’s question, and have included additional experiments for $r$ = 8 and $r$ = 32 with the LLaMA model on the math datasets. The new results are reported in Appendix D.
>
> As shown in the results, our method maintains the same trends as the baseline across different rank configurations.
>
> ### Weakness 2
> >  The LoRA adapters are trained following the QLoRA setup. It would be interesting to evaluate whether the proposed method generalizes to LoRA adapters trained on full-precision base models.
>
> Thank you for raising the question. We chose to train using QLoRA since training these models in full precision was not feasible due to our limited GPU budget. QLoRA is pretty standard in large model training nowadays and has been shown to have minimal effect on performance. To address the reviewer’s comment, we have trained llama2_7b on the math dataset (for two epoch) in full precision and include the result of evaluating on GSM8k:
>
> | Model | FP | 2@0.8 | 2@0.9 | 3@0.8 | 3@0.9 |
> |-----------------|-------|----------|----------|----------|----------|
> | QLoRA training | 58.53 | 51.25 | 52.16 | 53.6 | 56.86 |
> | fp training | 59.28 | 51.18 | 51.40 | 56.10 | 59.06 |
>
> As it can be seen, training either in full precision or using QLoRA has minimal effect on the results in the downstream task. Thus, we chose QLoRA for efficiency considerations. We explained our design choice in a footnote on Page 8.
> ### Change 1
> > How sensitive is performance to the LoRA rank (e.g., r=8, r=32)? Does the method degrade when the singular spectrum is flatter?
>
> The method is not sensitive to the rank, which is shown by our additional experiments with r=8 and r=32 in Appendix D. Thanks!
> ### Change 2
> > How does LoraQuant perform when applied to LoRA trained in full precision (without QLoRA base model)?
>
> We have included results for training in full precision to show our method still maintains its performance.

---

### Review · Reviewer_o9CR · 2026-04-16

**Summary Of Contributions:**

This work presents a post-training quantization scheme for LoRA adapters, the authors propose to identify high-importance and low-importance components from SVD, high and low bit-widths are assigned accordingly. Evaluations span LLaMA2-7B/13B and Mistral-7B across reasoning, coding, and summarization tasks.

Overall, the paper is easy to follow, the SVD-based importance ranking followed by mixed-precision assignment is intuitive and explained in a straightforward manner.

**Additional Comments:**

My previous review comments are also attached here.

**Audience:**

Yes

**Audience Explanation:**

The authors propose to identify high-importance and low-importance components from SVD, high and low bit-widths are assigned accordingly. This importance-aware scheme for aligning bits would be interesting to the LLM quantization literature.

**Broader Impact Concerns:**

SVD-based importance ranking and mixed-precision quantization are both well-established techniques in the model compression literature. While their combination applied to LoRA matrices is a reasonable contribution, the overall technical novelty remains limited without new insights specific to the LoRA setting.

**Claims And Evidence:**

Yes

**Claims Explanation:**

SVD-based importance ranking and mixed-precision quantization are both well-established techniques in the model compression literature. While their combination applied to LoRA matrices is a reasonable contribution, the overall technical novelty remains limited without new insights specific to the LoRA setting.

**Requested Changes:**

A natural alternative is to directly fine-tune LoRA adapters in low precision from the start, rather than quantizing them after training. It would strengthen the paper to include a comparison with such quantization-aware training baselines, as they may achieve better accuracy at comparable compression levels.


The decision to partition singular components into exactly two groups appears somewhat arbitrary. Alternative configurations such as a three-way split with one high-precision and two distinct low-precision tiers are equally plausible. The authors should provide a justification or empirical analysis exploring why a two-way partition is preferred.

SVDQuant is absent from the experimental comparisons. Including this baseline is important to properly contextualize the contribution.

---

> ### Author Response · Authors · 2026-04-29
> **Response by Authors**
>
> We are glad that the reviewer recognizes the clarity of our paper and the intuitive motivation behind our method. Thanks! We address the reviewer’s concerns below.
>
> >the overall technical novelty remains limited without new insights specific to the LoRA setting.
>
>
>
> Our paper contains multiple contributions, including a novel setting of LoRA compression for LLM customization and a simple-yet-effective approach to mixed-precision quantization. As the reviewer has recognized, the “combination [of SVD and mixed-precision quantization] applied to LoRA matrices is a reasonable contribution”. Thank you!
>
> > A natural alternative is to directly fine-tune LoRA adapters in low precision from the start, rather than quantizing them after training. It would strengthen the paper to include a comparison with such quantization-aware training baselines, as they may achieve better accuracy at comparable compression levels.
>
>
> Our paper focuses on post-quantization in the ultra-low bit setting (e.g., 1-2 bits). Since quantization operations are non-differentiable, quantization-aware training typically relies on the straight-through estimator (STE) to approximate gradients during backpropagation. They generally require full model training from scratch and therefore incur substantial computational cost. Further, quantization-aware training becomes more challenging in the ultra-low bit setup, as it is essentially a discrete optimization problem and gradient-based methods typically fail. Therefore, we focus on post-quantization methods.
>
> We nevertheless thank the reviewer for the suggestion, and have included quantization-aware training in the Related Work section.
>
> >The decision to partition singular components into exactly two groups appears somewhat arbitrary. Alternative configurations such as a three-way split with one high-precision and two distinct low-precision tiers are equally plausible. The authors should provide a justification or empirical analysis exploring why a two-way partition is preferred.
>
>
> Thanks for the question. We chose a two-group partition, simply because it has already achieved our primary goal of ulta-low bit quantization (avg. <2 bits). That being said, our approach can be easily extended to more sub-LoRAs with different bitwidths (which is useful when the Transformer scales further in the future). We include a pilot experiment here to show that we can easily use a higher number of partitions to get better results.
>
> | Method                    | GSM8K | Avg_bit |
> |---------------------------|-------|---------|
> | LoRAQuant 2@0.8          |    51.25   | 1.65     |
> | LoRAQuant 2@0.9          |    52.16   | 1.82    |
> | LoRAQuant 3@0.1 2@0.7    |    52.43   |    1.78     |
> | LoRAQuant 3@0.2 2@0.6    |     52.47  |     1.78    |
>
> The last two rows here are mixed quantization with three partitions. LoRAQuant 3@$x_1$ 2@$x_2$ means that the first $x_1\times 100\%$ percent of singular value components are stored in 3 bits, the next $x_2\times 100\%$ percent are in 2 bits, and the remaining values in 1 bit. Results show that by using 3 partition and keeping some parts of the LoRAs on higher precision we are able to achieve better performance with slightly higher average bit which is expected. We have added a footnote on page 3 discussing this.
>
> >SVDQuant is absent from the experimental comparisons. Including this baseline is important to properly contextualize the contribution.
>
> Thanks for the question. In the original submission we have an analysis in Appendix D (now Appendix E), showing that SVDQuant is not competitive with other baselines under the ultra-low-precision setting. In the revision, we have revised the text for clarification.

---

### Decision · Action_Editor_YSRF · 2026-06-03

**Recommendation:** Accept as is

**Additional Comments:**

The reviewers were generally positive about the paper and found the authors responsive to the concerns raised during the discussion phase. Although I recommend acceptance as is, the authors may consider the following optional improvements for the final version:
1. Consider clarifying the distinction between LoRAQuant and SVDQuant earlier in the paper, as this was a recurring point of discussion during the review process.
2. Consider expanding the discussion of the practical deployment scenario and the expected scale of LoRA adapter collections in real-world systems to further motivate the significance of the problem.
3. Consider incorporating some of the additional experimental observations from the author response (e.g., results on different LoRA ranks and full-precision LoRA training) into the main text or appendix for completeness.
4. Consider adding a brief discussion on potential extensions beyond the current two-part partitioning scheme, as the additional experiments suggest that multi-level precision allocation is a natural future direction.

**Audience:**

Yes

**Audience Explanation:**

LoRA-based fine-tuning has become a standard approach for adapting large language models, and efficient storage and deployment of large collections of LoRA adapters is an increasingly important practical challenge. The reviewing team agrees that the findings of this paper are of interest to researchers and practitioners working on parameter-efficient fine-tuning, model compression, quantization, and efficient LLM deployment.

**Claims And Evidence:**

Yes

**Claims Explanation:**

This work studies the problem of ultra-low-bit quantization for LoRA adapters, motivated by deployment scenarios where a large number of LoRA modules must be stored or loaded simultaneously. The paper proposes LoRAQuant, a mixed-precision post-training quantization framework that first applies SVD to reparameterize LoRA adapters and then allocates different bit-widths to components of different importance.

The reviewers agree that the paper addresses a practically relevant and timely problem. The proposed approach is simple, intuitive, and supported by extensive empirical evaluation across multiple model families and downstream tasks, including mathematical reasoning, code generation, and summarization. The experiments consistently demonstrate that LoRAQuant achieves substantially lower average bit-widths while maintaining competitive performance compared to existing quantization baselines.

Several questions were raised during the review process regarding the comparison to SVDQuant, the choice of a two-part partitioning strategy, robustness across different LoRA ranks, applicability beyond the QLoRA setting, and the role of the proposed quantization reconstruction procedure. The authors provided detailed responses and additional experiments addressing these concerns. Subsequent reviewer recommendations were positive, and one reviewer explicitly indicated that their questions had been addressed while maintaining a concern about methodological novelty.

Overall, the reviews support the view that the work addresses a relevant and practically important problem in LoRA compression, although the degree of methodological novelty was viewed as limited by some reviewers.